

# Prognostic accuracy of preoperative nutritional indicators of survival in head and neck cancer patients

Neyara dos Santos Oliveira[1], Tercio Guimarães Reis[2], Milena Souza Freitas[3], Marluce Matos Macêdo[3], Jean Carlos Zambrano Contreras[1], Márcio Campos Oliveira[4] and José Bessa Júnior[4]

[1] Postgraduate Program in Public Health, State University of Feira de Santana (UEFS), Feira de Santana, Bahia, Brazil
[2] Santa Casa de Misericórdia de Feira de Santana, Feira de Santana, Bahia, Brazil
[3] Head and Neck Cancer Research Group, Feira de Santana, Bahia, Brazil
[4] Department of Health, State University of Feira de Santana (UEFS), Feira de Santana, Bahia, Brazil

## ABSTRACT

Head and neck cancer (HNC) patients are known to have high nutritional risk and a high prevalence of malnutrition. The diagnosis of HNC, together with sequelae and other consequences of cancer treatment, directly impacts survival.

**Aim**. To determine overall 5-year survival in HNC patients submitted to surgery as their initial treatment and to assess the prognostic accuracy of nutritional anthropometric measurements weight loss percentage (%WL), body mass index (BMI), triceps skinfold (TSF), adductor pollicis muscle thickness (APMT), and calf circumference (CC) to predict survival in this population.

**Methods**. A prospective cohort study of HNC patients treated at a cancer referral center in Bahia's countryside was conducted.

**Results**. Seventy-eight patients were included in this study and were followed up for a 5-year period, with an overall survival rate of 52.1%. Most patients were male (83.3%), with a median age of 65.5 years [55–72 years], and most had low education levels, low household income, and a lifestyle marked by alcohol drinking and tobacco smoking. Median values of all nutritional anthropometric variables assessed in this study were significantly lower among patients who died during follow-up, suggesting greater impairment of nutritional status in this group. All nutritional indicators were found to be predictors of survival in the study population, with a prognostic accuracy of 74% for TSF 95% CI [0.63–0.83], 68% for BMI 95% CI [0.56–0.78], 65% for CC 95% CI [0.53–0.75], 63% for APMT 95% CI [0.51–0.63], and 63% for %WL 95% CI [0.51–0.73].

**Conclusion**. The overall 5-year survival rate was found to be 52.1%, and all nutritional anthropometric variables, namely BMI, %WL, TSF, APMT, and CC, were found to be good predictors of survival in HNC patients initially treated with surgery.

Corresponding author
Neyara dos Santos Oliveira,
neyara.o@gmail.com

## INTRODUCTION

Head and neck cancer (HNC) patients are known to have high nutritional risk and a high prevalence of malnutrition. Multiple diagnostic criteria are employed to detect malnutrition, whose prevalence among HNC patients ranges from 19% to 70%, depending on the diagnostic criteria adopted (*Reis, 2014*; *Oliveira, 2018*; *Kubrak et al., 2020*).

In addition to the degree of nutritional impairment often present at the time of diagnosis, the adverse effects of cancer treatment add to the deterioration of nutritional status. Changes in mental health and quality of life facing these patients also directly impact on their life expectancy (*Oliveira, 2018*).

Despite its curative potential, treatment-related sequelae of HNC surgery may include orofacial deformities and oropharyngeal dysphagia, which result from partial or complete damage to anatomical structures directly affected by the tumor or adjacent to it, thus leading to reduced food intake and negative changes in patients' eating habits. As a result, nutritional disorders such as malnutrition and dehydration may arise, as well as impaired resistance to infection, culminating in unfavorable clinical outcomes (*Bento, 2017*; *Melgaço, Vicente & Gama, 2021*).

The concept of survival transcends the number of days following diagnosis and treatment. More broadly, it encompasses physical, psychosocial, and economic aspects, beyond diagnostic and therapeutic stages (*Simcock & Simo, 2016*).

Weight loss and malnutrition are well-established, independent risk factors for postoperative complications in HNC (*Oliveira, 2018*; *Caburet et al., 2020*), and there are many tools for assessing nutritional status. Anthropometric measurements and indicators, including weight loss percentage (%WL), body mass index (BMI), triceps skinfold (TSF), adductor pollicis muscle thickness (APMT), and calf circumference (CC), are inexpensive and easy to obtain. However, few studies have looked into the predictive role of these preoperative indicators in survival rates of surgically treated HNC patients (*Datema, Ferrier & De Jong, 2011*; *Kubrak et al., 2020*; *Santos, Viani & Pavoni, 2021*).

In spite of the large number of measurements and indicators of nutritional status being used for the diagnosis of malnutrition, it remains controversial which of them has the best prognostic accuracy to predict treatment outcomes and survival in different clinical settings.

The aim of this project was therefore to assess overall 5-year survival among patients diagnosed with head and neck squamous cell carcinoma initially treated with surgery and to evaluate the prognostic accuracy of prespecified nutritional anthropometric variables (BMI, %WL, TSF, APMT, and CC) to predict survival in this population.

## METHODS

All eligible patients were consecutively included in the study between November 2016 and November 2022. Men and women aged 18 years or older who were submitted to a bloc surgery as the initial treatment of T2 to T4 (*Amin et al., 2018*) cancer of the oral cavity, larynx, oropharynx, hypopharynx, or nasopharynx were included in this study.

Data were collected as previously described in *Oliveira et al. (2024)*. The method relating to the object of study of this manuscript is specified below.

Patients diagnosed with dyslipidemia and taking lipid-lowering drugs and glucocorticoids, those with diseases affecting the normal metabolism of hepatic proteins, such as nephrotic syndrome, congestive heart failure, and cirrhosis, and individuals who did not provide written informed consent were excluded from the study.

This project was approved by the Human Research Ethics Committee of the State University of Feira de Santana under the protocol no. 1.399.962.

Eligible subjects were referred to an experienced dietitian by the surgeon the week before surgery. During this visit, patients provided written informed consent and had their sociodemographic and clinical data collected, as well as their nutritional status evaluated.

Body weight was measured with a *Welmy*® mechanical scale, with a 150-kg capacity and an accuracy of 100 g, while height was measured with an attached stadiometer with a maximum measuring capacity of 2.05 m, conforming to the rules described by *Lohman, Roche & Martorell (1988)*. BMI was derived from Quételet's equation [BMI = weight/(height$^2$)].

Triceps skinfold (TSF), weight loss percentage (%WL), adductor pollicis muscle thickness (APMT), and calf circumference (CC) were calculated and utilized as secondary indicators of nutritional status. %WL was calculated as follows: [(usual weight - actual weight)/(usual weight) x 100] (*Blackurn & Bistrian, 1977*). A Lange skinfold caliper was used to measure TSF and APMT and was calibrated according to techniques described by *Lohman, Roche & Martorell (1988)* and *Lameu et al. (2004)*, respectively. A Seca inelastic fiberglass measuring tape was used to measure CC to the nearest 0.1 cm according to a previously published technique (*Lee & Nieman, 1995*).

All surgical procedures were carried out by the head and neck surgeon from the Head and Neck Cancer Research Group (NUPESCAP) at the HDPA. Data on the types of surgery and simultaneous neck dissection, the need for flap reconstruction, and the occurrence of intraoperative complications were collected during surgery.

All patients received nutrition support therapy consisting of an industrialized formula was initiated enterally up to 12 h after surgery as per protocol and patient specificities. During their hospital stay, patients were seen daily by the surgeon responsible for the procedure in order to assess patient condition and monitor postoperative complications.

After discharge, patients were followed up by a multidisciplinary team, including the responsible medical team, and when necessary were referred for adjuvant treatments (radio and chemotherapy). During follow-up, patients were seen by dietitians in order to evaluate their nutritional status and to provide education on tube feeding and changes in diet such as dietary supplementation and diet texture modifications.

Death (date of death) regardless of cause was the event of interest for overall survival analysis. The date of surgery was defined as the start of follow-up for each participant. Patients who remained alive until the date of last follow-up were censored. Brazil's National Civil Registry Information System was searched online for patients' names if they were lost to follow-up at https://www.registrocivil.org.br/ (*TJBA, 2023*). Five-year survival analysis was conducted using a time-to-event approach, and all patients were followed from the date

of surgery to either the date of death or censoring on February 28, 2023. Therefore, not all patients reached five years of follow-up, but their partial follow-up data were incorporated in the Cox regression model, which accounts for varying follow-up durations.

Microsoft Excel was used for tabulation of data and GraphPad Prism, version 10.0.0 for Windows (San Diego, CA, USA), was used for statistical analysis. Continuous and ordinal quantitative variables were presented as medians and interquartile ranges, while qualitative variables were expressed as absolute values and proportions.

The Student's $t$-test and Mann–Whitney $U$ test were used to compare continuous variables, while the chi-squared test and its variants were used to compare categorical variables. We adopted $p$ values less than 0.05 ($p < 0.05$) to indicate statistical significance.

Kaplan–Meier curves were used to estimate overall survival probability over time, and 95% confidence intervals were used as a measure of precision.

Receiver operating characteristic (ROC) curves were utilized to compare and calculate the overall accuracy of the anthropometric indicators, which was measured by the area under the ROC curve.

We performed both univariate and multivariate Cox proportional hazards regression models to evaluate the relationship between preoperative nutritional anthropometric variables (BMI, TSF, %WL, CC and APMT) and 5-year overall survival in patients diagnosed with head and neck squamous cell carcinoma initially treated with surgery. The predictors included in the model were BMI, CC, TSF, AMPT, and %WL. The survival function was modeled using the *coxph* function from the survival package in R, with the formulation *Surv (survival_months, death)* as the dependent variable. The multivariate model included adjustments for potential confounders such as age, sex, clinical stage (stage II area excluded duo to small number of patients and stage III is considered reference category), and treatment modality.

Adjusted risk values for each individual were then predicted using the predict function with the "risk" type. The model's predictive capacity was assessed through the ROC curve generated by the ROC function from the *pROC package*.

Additionally, the proportional hazards assumption was tested using the *cox.zph* function, which confirmed that the assumption of proportionality was not violated for any individual predictors or the model as a whole ($p > 0.05$ for all predictors).

## RESULTS

A total of 78 subjects were included in this study, with a median age of 65.5 years (55–72 years). Most of them were male and had low education levels, low income, and a lifestyle marked by alcohol drinking and tobacco smoking.

Sociodemographic and lifestyle characteristics were not significantly different between groups (alive *vs.* dead) in univariate analysis. These data are shown in Table 1.

Patients were followed up for up to 60 months. In this period, 40 deaths were recorded (51.2%), with a median survival time of 39 months (Fig. 1).

Most patients had cancer of the larynx (51.4%) and the oral cavity (44.8%). The most commonly observed primary tumor subsites were the glottis, in 48.7% of cases, the floor of

**Table 1  Sociodemographic and lifestyle characteristics of the study population.**

| Variable | Total (n = 78) | Alive (n = 38) | Dead (n = 40) | p value |
|---|---|---|---|---|
| | n (%) | n (%) | n (%) | |
| Age (years)* | 65.5 [55–72] | 63.0 [53.7–67.2] | 67.5 [57.5–72.0] | 0.06 |
| Sex | | | | |
| Male | 65 (83.3%) | 29 (76.3%) | 36 (90.0%) | 0.17 |
| Female | 13 (16.7%) | 9 (23.7%) | 4 (10.0%) | |
| Education level | | | | |
| Illiterate | 17 (21.8%) | 6 (15.8%) | 11 (27.5%) | 0.40 |
| Primary education | 49 (62.8%) | 26 (68.5%) | 23 (57.5%) | |
| Secondary education | 6 (7.7%) | 4 (10.5%) | 2 (5.0%) | |
| Higher education | 6 (7.7%) | 2 (5.2%) | 4 (10.0%) | |
| Household income | | | | |
| No income | 15 (19.2%) | 6 (15.8%) | 9 (22.5%) | 0.30 |
| Up to the minimum wage | 47 (60.2%) | 23 (60.5%) | 24 (60.0%) | |
| 1 to 2 times the minimum wage | 13 (16.8%) | 6 (15.8%) | 7 (17.5%) | |
| ≥ 2 times the minimum wage | 3 (3.8%) | 3 (7.9%) | 0 (0.0%) | |
| Alcohol drinking | | | | |
| No | 7 (9.0%) | 2 (5.3%) | 5 (12.5%) | 0.16 |
| Yes | 71 (91.0%) | 36 (94.7%) | 35 (87.5%) | |
| Tobacco smoking | | | | |
| No | 10 (12.8%) | 4 (10.5%) | 6 (10.0%) | 0.48 |
| Yes | 68 (87.2%) | 34 (89.5%) | 34 (90.0%) | |

**Notes.**
*Median [interquartile range].

the mouth (24.3%), the tongue (12.8%), and the base of the tongue (7.7%). As for clinical staging, 66.6% of patients had advanced-stage disease (stages III and IV), and these patients were overrepresented among those who died. Patient clinical characteristics are detailed in Table 2.

Nutritional aspects of the study population in both groups (alive and dead patients) are presented below. All study variables were significantly different between groups in that they had lower median values (greater impairment) among those who died (Table 3).

ROC curves demonstrating the prognostic accuracy of the anthropometric indicators to predict survival in the HNC patients included in this study are shown in Fig. 2. We found a discriminatory power (accuracy) to predict survival of 74% for TSF 95% CI [0.63–0.83], of 68% for BMI 95% CI [0.56–0.78], of 65% for CC 95% CI [0.53–0.75], of 63% for APMT 95% CI [0.51–0.63], and of 63% for %WL 95% CI [0.51–0.73].

The results of the Cox model analysis demonstrated that TSF was the only significant predictor of survival among the variables analyzed. A higher TSF was associated with a reduced hazard of death, with a hazard ratio (HR) of 0.88 (95% CI [0.79–0.98], $p = 0.022$)

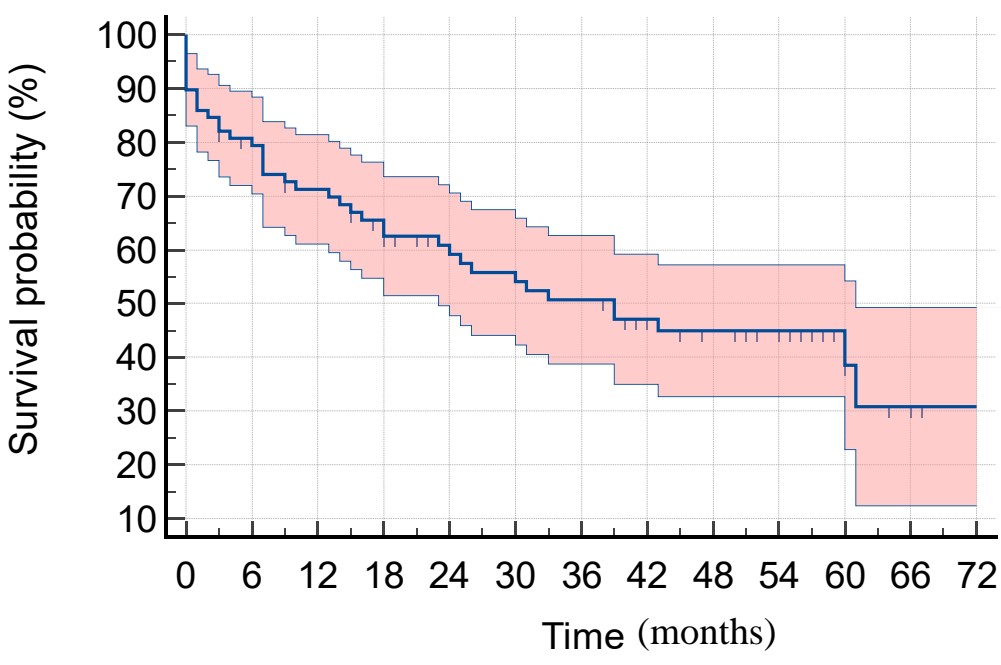

**Figure 1** Five-year survival in HNC patients submitted to surgery as their initial treatment.

(Table 4). The ROC curve for TSF indicated moderate predictive performance for the occurrence of death, as evidenced by the area under the curve (AUC).

In contrast, the other variables—BMI, CC, AMPT, and %WL—did not show statistical significance in predicting survival. Specifically, BMI had an HR of 0.92 (95% CI [0.79–1.08], $p = 0.3$), CC had an HR of 0.93 (95% CI [0.81–1.07], $p = 0.3$), AMPT had an HR of 0.96 (95% CI [0.87–1.05], $p = 0.3$), and %WL was similarly non-significant (Table 4).

Although clinical stage was more frequently advanced among deceased patients, it did not reach statistical significance in univariate analysis. However, we recognize the well-established prognostic importance of staging. Even after the adjustment for confounders variable, triceps skinfold thickness (TSF) remained independently associated with mortality.

## DISCUSSION

The overall 5-year survival rate was found to be 52.1% in the study population, with a median survival time of 39 months. This rate is higher than that reported in a Danish study, which was 47.8% 95% CI [47.1–48.5] over a 5-year period after the diagnosis of HNC (*Jakobsen et al., 2018*).

The overall 5-year survival rates after the diagnosis of HNC range from 30 to 60% in studies conducted in Brazil. In a retrospective cohort study carried out at the Oncocentro

**Table 2  Preoperative clinical characteristics of the study population.**

| Variable | Total (n = 78) | Alive (n = 38) | Dead (n = 40) | p value |
|---|---|---|---|---|
| | n (%) | n (%) | n (%) | |
| **Primary tumor site** | | | | |
| Oral cavity | 35 (44.8%) | 16 (42.1%) | 19 (47.5%) | 0.19 |
| Oropharynx | 3 (3.8%) | 3 (7.9%) | 0 (0.0%) | |
| Larynx | 40 (51.4%) | 19 (50.0%) | 21 (52.5%) | |
| **Primary tumor subsite** | | | | |
| Tongue | 10 (12.8%) | 4 (10.5%) | 6 (15.0%) | 0.20 |
| Buccal mucosa | 2 (2.6%) | 0 (0.0%) | 2 (5.0%) | |
| Floor of the mouth | 19 (24.3%) | 9 (23.7%) | 10 (25.0%) | |
| Lower gingiva | 1 (1.3%) | 1 (2.6%) | 0 (0.0%) | |
| Base of the tongue | 6 (7.7%) | 5 (13.1%) | 1 (2.5%) | |
| Aryepiglottic fold | 2 (2.6%) | 2 (5.2%) | 0 (0.0%) | |
| Glottis | 38 (48.7%) | 17 (44.8%) | 21 (52.5%) | |
| **Clinical stage** | | | | |
| I | 2 (2.6%) | 2 (5.2%) | 0 (0.0%) | 0.020 |
| II | 24 (30.8%) | 16 (42.1%) | 8 (20.0%) | |
| III | 17 (21.8%) | 6 (15.8%) | 11(27.5%) | |
| IV | 35 (44.8%) | 14 (36.8%) | 21 (52.5%) | |

**Table 3  Cox proportional hazards regression model to evaluate the relationship between preoperative nutritional anthropometric variables and 5-year overall survival in patients (adjusted).**

| Variable | Death | | | Time to death | | |
|---|---|---|---|---|---|---|
| | OR[I] | 95% CI[I] | p-value | HR[I] | 95% CI[I] | p-value |
| TSF | 0.80 | 0.66, 0.93 | 0.009 | 0.88 | 0.79, 0.98 | 0.022 |
| BMI | 1.08 | 0.84, 1.40 | 0.6 | 0.92 | 0.79, 1.08 | 0.3 |
| CC | 0.93 | 0.73, 1.16 | 0.5 | 0.93 | 0.81, 1.07 | 0.3 |
| AMPT | 0.90 | 0.77, 1.04 | 0.2 | 0.96 | 0.87, 1.05 | 0.3 |
| Clinical stage | | | | | | |
| II | – | – | – | – | – | – |
| III | 2,04 | 0,48, 9,09 | 0,3 | 2,12 | 0,83, 5,36 | 0,1 |
| IV | 1,23 | 0,33, 4,48 | 0,7 | 1,31 | 0,56, 3,06 | 0,5 |

**Notes.**

[I] OR, Odds Ratio; CI, Confidence Interval; HR, Hazard Ratio.

[II] Triceps Skinfold Thickness (TSF), Body Mass Index (BMI), Calf Circumference (CC), Adductor Pollicis Muscle Thickness (AMPT).

de São Paulo, the overall 5-year survival rate was found to be 35% regardless of the modality of the initial treatment (*Santos, Viani & Pavoni, 2021*). The overall survival rate for mouth cancer was reported to be 32.2% in a study by Bonfante and colleagues (*Bonfante et al., 2014*). As for laryngeal cancer, the overall 3-year survival rate was found to be 58% for patients with stage III disease and 42% for those with stage IV cancer (*Karlsson et al., 2014*).

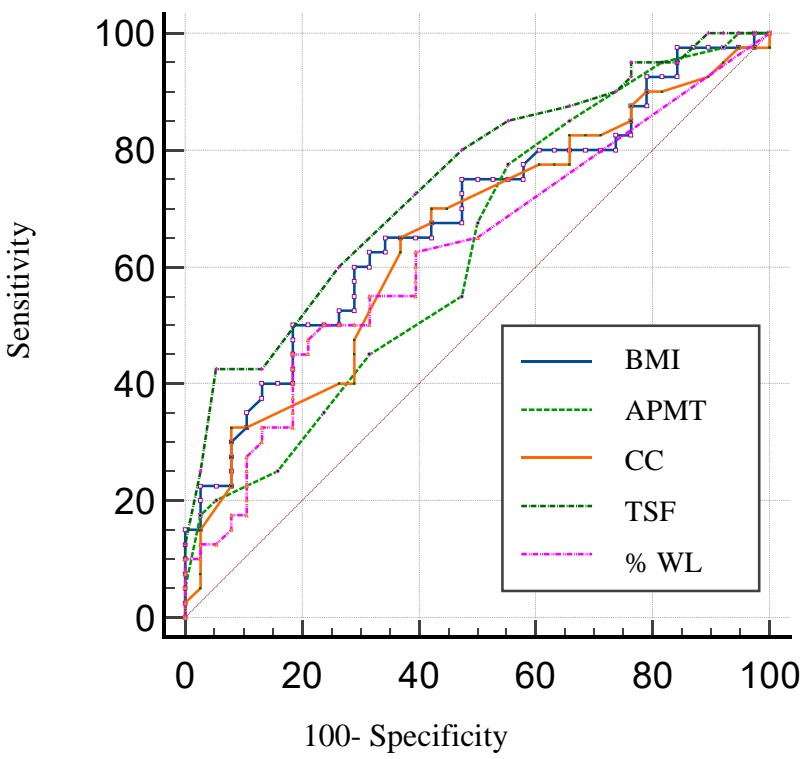

**Figure 2** ROC curves showing the prognostic accuracy of the anthropometric indicators.

**Table 4** Correlation between the anthropometric indicators and death in the study population.

| Variable | Total (n = 78) | Alive (n = 38) | Dead (n = 40) | p value |
|---|---|---|---|---|
| | Mdn [IC$_{25}$–IC$_{75}$] | Mdn [IC$_{25}$–IC$_{75}$] | Mdn [IC$_{25}$–IC$_{75}$] | |
| BMI (kg/m$^2$) | 23.2 [20.5–26.2] | 24.3 [22.2–26.1] | 22.1 [19.4–24.6] | <0.01 |
| % WL | 1.53 [0–6.3] | 0.5 [0–3.2] | 2.8 [0–7.4] | 0.02 |
| TSF (mm) | 9.0 [6.7–12.0] | 11.5 [8.7–14] | 8.0 [5.2–10.0] | <0.001 |
| APMT (mm) | 10.0 [8–13.3] | 11.5 [8.7–14] | 10.0 [7.2–12] | 0.03 |
| CC (cm) | 33.3 [32–35.8] | 35.0 [32.0–37.1] | 33.0 [31.0–35.0] | 0.03 |

**Notes.**
Mdn, median.

The epidemiology of the study population was characterized by the predominance of males, of advanced age, with low income and low education levels, and of alcohol drinkers, which is in line with the HNC literature (*Carvalho, 2017*; *Brasil, 2018*; *Brasil, 2020*; *Brasil, 2022*; *Silva et al., 2020*).

Cancers of the oral cavity and the larynx figure among the 10 most frequent types of cancer in Brazil and are thus considered a serious public health issue (*Silva et al., 2020*).

In this study, 66.6% of patients had advanced-stage disease (stages III and IV) at the time treatment was initiated, which is common in HNC (*Ingarfield et al., 2019*; *Santos*

*et al., 2022*; *Bonfante et al., 2014*). Early diagnosis of HNC is not difficult, but patient delay in help-seeking, medical malpractice in diagnosis/follow-up, and lack of appropriate treatment facilities likely lead to late diagnosis in most cases (*Huang, Imam & Nguyen, 2020*).

Data from Brazil's National Cancer Institute (2020) (*Brasil, 2020*) showed the time between diagnosis and first treatment of lip and oral cavity cancers was over 60 days in 2018 and 2019 for most cases from the North, Northeast, and Southeast regions of Brazil. This clearly indicates healthcare services are facing challenges in follow-up care, which may be the reason for the advanced stage treatment initiation in this study.

Diagnostic delay favors tumor growth, infiltration of adjacent tissues, and metastatic dissemination, thus hindering treatment and warranting more aggressive therapies with greater impact on quality of life (*Yan, Agrawal & Gooi, 2018*; *Santos et al., 2022*). This explains the positive association between late-stage cancer and death ($p = 0.020$).

Early diagnosis of HNC is associated with better treatment outcomes. For this purpose, it is essential to ensure the immediate initiation of cancer treatment aiming to reducing mortality and maintain quality of life. Evidence shows HNC treatments are improving against a background of increasing incidence (*Abbott et al., 2016*; *Simcock & Simo, 2016*).

It is worth remembering that data collection for this study happened during the pandemic period. And the additional complications caused by the COVID-19 pandemic, including the need for social isolation and the overburdened healthcare services, which have led HNC patients, an already vulnerable population, to experience treatment delays and interruptions, surely contributing to an increased frequency of advanced cancer and reduced survival among these patients. A 20% increase in mortality from HNC is estimated to occur as a result of the COVID-19 pandemic (*Huang, Imam & Nguyen, 2020*; *Chone, 2021*).

As for the nutritional aspects, our population was found to have BMI values bordering on malnutrition for older adults (23.2 kg/m$^2$) (*Pan American Health Organization, 2002*), with a median TSF value of 9.0 mm, which is consistent with a state of severely depleted fat reserves (TSF <5th percentile) (*Frisancho, 1990*), and median APMT (10 mm) (*Lameu et al., 2004*) and CC (33 cm) (*Cruz-Jentoft et al., 2019*) values indicating normal muscle reserve.

These findings are in line with the literature in that malnutrition is considered common in HNC, with a prevalence ranging from 40 to 80% depending on the diagnostic criteria adopted, as well as on the type and location of the tumor (*Karnell et al., 2015*; *Talwar et al., 2016*).

Individuals with HNC have reduced food intake prior to treatment due to their lifestyle of alcohol consumption and smoking, the location of the neoplasm, and the catabolic properties of the tumor itself. In addition, HNC therapy has debilitating physical effects on organs essential for normal human activities, such as breathing, communication, and nutrition and hydration, resulting from the need to use a feeding tube and/or tracheostomy, dysphagia, trismus, mucositis, among others (*Bento, 2017*; *Melgaço, Vicente & Gama, 2021*). The complex interaction between the anatomical structures invaded by the tumor and the

effects of the treatments, as well as the operational and financial difficulties following the prescribed diet, further contribute to the deterioration of the nutritional status.

All nutritional variables assessed in the present study were shown to be good predictors of survival after HNC surgery. There are few published studies evaluating the role of these nutritional variables in predicting survival among HNC patients. In a previously published work of ours, nutritional anthropometric variables (BMI, TSF, APMT, and CC) were found to have good accuracy to predict severe postoperative complications (*Oliveira et al., 2022*).

Out of the indicators assessed in the present study, BMI has been more consistently positively associated with survival in HNC (*Liu et al., 2006*; *Karnell et al., 2015*; *Takenaka et al., 2014*); *Gama et al. (2017)* demonstrated underweight HNC patients (BMI < 18.5 kg/m$^2$) to be nearly twice as likely to die (HR: 1.89; 95% CI [1.2–3.1]) when compared to those with normal weight. Being overweight at diagnosis, on the other hand, was found to be a protective factor, resulting in improved survival (HR: 0.55; 95% CI [0.4–0.8]).

Weight loss is easily observed in HNC patients and it is a good indicator of the difficulties facing these patients with regard to maintaining an adequate nutritional status in the face of multiple metabolic changes imposed by both the physical presence of the tumor and the consequent negative impact it has on food intake. This variable has been previously found to be associated with postoperative complications such as wound dehiscence, surgical wound infection, and fistula formation (RR: 2.0; $p < 0.001$), as well as acute cardiovascular events, pneumonia, kidney failure, and sepsis (RR: 2.6; $p < 0.001$), in surgical HNC patients (*Gourin, Couch & Johnson, 2014*).

In a survival analysis study, all degrees of weight loss were found to be associated with worse survival after controlling for age, clinical staging, and tumor location (*Kubrak et al., 2020*). An inverse relationship between the probability of 5-year survival for patients with head and neck epidermoid carcinoma and the increase in %WL ($p < 0.01$) was reported in a study by *Datema, Ferrier & De Jong (2011)*. In line with these findings, %WL was shown to be a good predictor of survival in HNC patients in the present study.

TSF was predictor independent and yielded an accuracy of 74% to predict survival in this study. This indicator is a simple and noninvasive tool for assessing fat reserves. TSF was reported to have similar accuracy (75%) to predict severe postoperative complications in HNC patients in a study by *Oliveira (2018)* (95% CI [0.56–0.94]; $p = 0.031$).

APMT is a cheap and rapid tool for estimating loss of muscle mass (*Bragagnolo et al., 2009*) and has been used in multiple clinical settings, with significant results in nutrition diagnosis and in prognostic analysis of in-hospital postoperative complication and length of hospital stay (*Andrade, Lameu & Luiz, 2005*; *Gonzalez et al., 2015*; *Rosca et al., 2015*). To our knowledge, this is the first study to assess the role of APMT in predicting survival in HNC, yielding an accuracy of 63%.

Both the cancer itself and the onset of malnutrition may cause patients to reduce daily work activities and may potentially induce catabolism, resulting in a progressive decrease in APMT (*Caporossi et al., 2012*). In a cross-sectional study conducted by *Freitas et al. (2010)*, mean APMT was reported to be 13 ± 3.2 mm and was not found to be associated with length of hospital stay or mortality.

CC yielded an accuracy of 65% to predict survival in the present study (95% CI [0.53–0.75]; $p = 0.03$). Previous research has shown CC to be an independent predictor of severe postoperative complications (*Oliveira et al., 2022*) and survival in HNC (*Sousa et al., 2020*), gastric and colorectal cancers (*Sousa et al., 2022*), and palliative care (*Da Silva et al., 2019*). This indicator (CC < 31 cm) is considered by the European Consensus on Sarcopenia to be a predictor of physical performance and survival in older adults regardless of underlying diseases (*Mello, Waisberg & Silva, 2016*; *Cruz-Jentoft et al., 2019*).

Our findings demonstrate the prognostic value of TSF even after adjusting for key clinical confounders, including stage, age, and sex. This supports TSF as a nutritional predictor of survival in head and neck cancer. Moreover, our results corroborate with evidence linking nutritional status to sarcopenia. Recent studies have shown that preoperative sarcopenia is an independent predictor of poor outcomes in head and neck cancer highlighting the need for early identification and intervention to prevent muscle mass depletion in head and neck cancers (*Erul et al., 2023*).

There are many factors involved in the mortality and survival of individuals with HNC. We demonstrated that preoperative nutritional status is positively associated with mortality and survival at 5 years of follow-up. However, we know that there are several other factors involved in the survival of these subjects, which are not the focus of discussion in this manuscript, such as: complications from surgery, need for adjuvant therapy, other cancerous diseases, quality of medical and family care, among others.

The limitations of this study include its single-center design and its limited sample size. However, we consider our results to be sufficient to suggest the application of these indicators in clinical practice can easily identify patients at risk of death, who may benefit from targeted care in order to preserve nutritional status and consequently improve prognosis. Furthermore, it also provides additional information to help the medical team plan therapeutic care for these cases.

## CONCLUSION

The overall 5-year survival rate was found to be 47% among HNC patients submitted to surgery as their initial treatment. The nutritional anthropometric variables (TSF, BMI, %WL, CC, and APMT) showed moderate accuracy in predicting survival (63 a 74%). Only the TSF remained associated with mortality after multivariate adjustment.

### Funding
The authors received no funding for this work.

### Competing Interests
The authors declare there are no competing interests.

## Author Contributions

- Neyara dos Santos Oliveira conceived and designed the experiments, performed the experiments, analyzed the data, prepared figures and/or tables, authored or reviewed drafts of the article, and approved the final draft.
- Tercio Guimarães Reis conceived and designed the experiments, performed the experiments, analyzed the data, authored or reviewed drafts of the article, and approved the final draft.
- Milena Souza Freitas performed the experiments, prepared figures and/or tables, and approved the final draft.
- Marluce Matos Macêdo performed the experiments, authored or reviewed drafts of the article, and approved the final draft.
- Jean Carlos Zambrano Contreras analyzed the data, prepared figures and/or tables, authored or reviewed drafts of the article, and approved the final draft.
- Márcio Campos Oliveira conceived and designed the experiments, analyzed the data, authored or reviewed drafts of the article, and approved the final draft.
- José Bessa Júnior conceived and designed the experiments, analyzed the data, prepared figures and/or tables, authored or reviewed drafts of the article, and approved the final draft.

## Human Ethics

The following information was supplied relating to ethical approvals (i.e., approving body and any reference numbers):

The Research Ethics Committee of the State University of Feira de Santana, Bahia, Brazil approved the study (The protocol no. 1.399.962).

## Data Availability

The raw measurements are available in the Supplementary File.

## Supplemental Information

Supplemental information for this article can be found online at http://dx.doi.org/10.7717/peerj.19496#supplemental-information.

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
