# Peer review of "Prognostic accuracy of preoperative nutritional indicators of survival in head and neck cancer patients"

_PeerJ, doi:10.7717/peerj.19496_

## Round 0.1 · original submission · Major Revisions

The authors are requested to carefully revise the manuscript and answer the questions raised by the reviewers. Serious concerns have been expressed regarding the study design - you must address these issues fully

·

Basic reporting

The study design and statistical methods have significant shortcomings. Classifying patients into groups based on survival status (alive vs. deceased) for predictive conclusions is not standard practice. Instead, ROC analysis should be conducted to determine sensitivity and specificity, followed by defining a cut-off value. Groups can then be compared based on this cut-off for overall survival (OS), progression-free survival (PFS), and treatment toxicity. Univariate analysis should first identify significant factors, which should then be included in a multivariate analysis to determine independent risk factors. This approach is more robust and reliable for identifying predictors of survival. The manuscript lacks a detailed survival analysis. Kaplan-Meier curves and multivariate analysis should be used to evaluate OS and PFS. The absence of multivariate analysis is a critical flaw.
The discussion includes points that are not presented as findings in the study. For example:
The manuscript mentions data from Brazil's National Cancer Institute about delays in treatment initiation, but it does not provide similar data from the current study. Why is this comparison included?
The impact of the COVID-19 pandemic on treatment delays and interruptions is discussed without presenting any corresponding results. What specific findings related to COVID-19 does this study report?
Abbreviations such as BMI, %WL, TSF, APMT, and CC should be spelled out the first time they are used to ensure clarity.

Experimental design

The study design and statistical methods have significant shortcomings. Classifying patients into groups based on survival status (alive vs. deceased) for predictive conclusions is not standard practice. Instead, ROC analysis should be conducted to determine sensitivity and specificity, followed by defining a cut-off value. Groups can then be compared based on this cut-off for overall survival (OS), progression-free survival (PFS), and treatment toxicity. Univariate analysis should first identify significant factors, which should then be included in a multivariate analysis to determine independent risk factors. This approach is more robust and reliable for identifying predictors of survival. The manuscript lacks a detailed survival analysis. Kaplan-Meier curves and multivariate analysis should be used to evaluate OS and PFS. The absence of multivariate analysis is a critical flaw.

Validity of the findings

The discussion includes points that are not presented as findings in the study. For example:
The manuscript mentions data from Brazil's National Cancer Institute about delays in treatment initiation, but it does not provide similar data from the current study. Why is this comparison included?
The impact of the COVID-19 pandemic on treatment delays and interruptions is discussed without presenting any corresponding results. What specific findings related to COVID-19 does this study report?
Abbreviations such as BMI, %WL, TSF, APMT, and CC should be spelled out the first time they are used to ensure clarity.

Reviewer 2 ·

Basic reporting

Congratulation to the group to focusing on possible nutritional factors as contributors to survival in patients with Head and neck cancer. Both the nutritional efforts as well as focus on factors limiting survival are relevant to investigate in this patient group.

There is a need for revision of language.

Literature references are used well. and background field is partly well covered.
However the authors do not discuss the possible causality of nutritional challenges for survival.
It is well described in the litterature that although an intensive effort is made to nourish the patients, both in hospital and after discharge, there is a great difference in the compliance of the patients and the practical as well as financial possibilities to comply with instructions. The authors describe the departments normal procedure- but fails to describe the scope of the effort.
The patients undergo large tumor ablations and reconstructive surgery in head and neck- treatments that often will be followed by radiotherapy. There is no registration of adjuvant treatments that could impact survival as well as impact the nutritional state.
Major surgery as described in this population often result in dysphagia and need for tube feeding - there is no description of either dysphagia or need for tube feeding.
The study only analyse overall survival- not disease specific survival- It is unclear whether the patients die due to cancerdisease, malnourishment or other causes.
Finaly- the observation that decreased nutritional status - as measured with BMI, antropometric measures- is related to decreased survival- what should the consequences of this be? Would it change the treatment decision?

The first time an abbreviation is used it should be written out completely- in this case: line 41: (BMI, %WL, TSF, APMT, and CC)

Experimental design

Method:
Observational design with 5 years follow-up post surgery.
78 cancer patients were operated over a 6 years period and followed for 5 years. However the latest patients could not be followed for 5 years as the inclusion period ended 2022 and observation/registration ended spring 2023. Median follow-up time is written- but it is unclear how many patients were not followed.
Well described exclusion criteria. Methods of measurements and calculations well described
Lack of data on adjuvant therapy
Lack of data on nutritional supplement- tube feeding, PEG, degree of dysphagia

In discussion section: line 192. It is not recommendable to compare a selected group of patients with registry data- change reference.

Validity of the findings

I would like to commend the authors for having focus on causes that may impact survival in this fragile patient group and on the well-chosen and applicable measures chosen to describe nutritional status.

However there are several possible other causalities that could impact survival- the authors do mention some- such as time to treatment. However other causes such as complication due to surgery, adjuvant therapy, other cancer diesases are not discussed

The data is provided- seems robust- however- survival data Kaplan Meier curve lack details on numbers at risk and censored patients
Table 1: Tobacco: inconsistency of numbers in regard to tobacco it says 10 pt is non smokers- and then 32 pt. in the “dead” category – the numbers do not fit with data sheet.

Additional comments

In general an important subject to analyse areas that may predict survival.
The methods used are good and the study well designed to show that nutritional impairment exist.

However the conclusion that nutritional state and anthropometric measures can be predictors of survival is an overestimate- i would only dare to conclude that nutritional state and measures are correlated with survival.

---

## Round 0.2 · Major Revisions

The authors are requested to carefully revise the manuscript and answer the questions raised by the reviewers.

·

Basic reporting

The manuscript is well-written and follows a clear and logical structure.

Experimental design

The experimental design is well-structured and aligns with the study objectives.

Validity of the findings

The manuscript presents relevant findings with potential clinical implications. The study design and methodology appear robust, supporting the validity of the results

Additional comments

The authors have addressed all my previous comments. As a minor suggestion, I recommend adding the reference PMID: 37042459 where relevant. Given the importance of nutritional markers, this study highlights the potential relationship between sarcopenia and these indicators. Additionally, it could contribute to the discussion on future directions for preventing sarcopenia in head and neck cancer patients.

Reviewer 3 ·

Basic reporting

1. The authors reported prognostic "accuracy" in the abstract and the whole text from ROC curves. As we know, the area under the ROC curve (AUC) is not equal to accuracy, although they are closely related.

Experimental design

1. Line110: All eligible patients were consecutively included in the study between November 2016 and
111 November 2022. and Line153: The end date for this analysis was February 28, 2023, and estimates were calculated for a 5-year follow-up period. How the authors handle patients who enter the cohort later in the study? Apparently, the longest follow-up time for patients entering the study in 2022 is less than 5 years. The author did not explain this clearly.
2. Univariate or multivariates Cox regression analysis was used? Any confounders were adjusted?

Validity of the findings

1. Although the authors supplemented the multivariate Cox regression analysis of nutritional indicators, they did not really consider the impact of other confounding factors and covariates (eg,. clinical stage) on survival. For example, a significant difference of clinical stage was observed in Table 2, and higher stage Ⅲ-Ⅳ proportion was found. However the authors did not attempt to control for the effect of stage on prognosis. I am afraid that a single nutritional indicators analysis is not enough to support the current conclusion.

Additional comments

None.

---

## Round 0.3 · accepted · Accept

After revisions, two reviewers agreed to publish the manuscript. I also reviewed the manuscript and found no obvious risks to publication. Therefore, I also approved the publication of this manuscript.

Reviewer 3 ·

Basic reporting

None

Experimental design

None

Validity of the findings

None

Additional comments

The authors have answered my confusion appropriately. Thank you.